# CoCo: Controllable Counterfactuals for Evaluating Dialogue State Trackers

**Shiyang Li**[*][†],  **Semih Yavuz**[*],  **Kazuma Hashimoto**,  **Jia Li**,  **Tong Niu**
**Nazneen Rajani**,  **Xifeng Yan**[†],  **Yingbo Zhou**,  **Caiming Xiong**
Salesforce Research        [†]University of California, Santa Barbara
```
{syavuz, k.hashimoto, jia.li, tniu, nazneen.rajani,
 yingbo.zhou, cxiong}@salesforce.com
{shiyangli, xyan}@ucsb.edu
```

## Abstract

Dialogue state trackers have made significant progress on benchmark datasets, but their generalization capability to novel and realistic scenarios beyond the held-out conversations is less understood. We propose controllable counterfactuals (CoCo) to bridge this gap and evaluate dialogue state tracking (DST) models on novel scenarios, i.e., would the system successfully tackle the request if the user responded differently but still consistently with the dialogue flow? CoCo leverages turn-level belief states as counterfactual conditionals to produce novel conversation scenarios in two steps: (i) *counterfactual goal* generation at turn-level by dropping and adding slots followed by replacing slot values, (ii) *counterfactual conversation* generation that is conditioned on (i) and consistent with the dialogue flow. Evaluating state-of-the-art DST models on MultiWOZ dataset with CoCo-generated counterfactuals results in a significant performance drop of up to 30.8% (from 49.4% to 18.6%) in absolute joint goal accuracy. In comparison, widely used techniques like paraphrasing only affect the accuracy by at most 2%. Human evaluations show that COCO-generated conversations perfectly reflect the underlying user goal with more than 95% accuracy and are as human-like as the original conversations, further strengthening its reliability and promise to be adopted as part of the robustness evaluation of DST models. [1]

## 1 Introduction

Task-oriented dialogue (TOD) systems have recently attracted growing attention and achieved substantial progress (Zhang et al., 2019b; Neelakantan et al., 2019; Peng et al., 2020; Wang et al., 2020b;a), partly made possible by the construction of large-scale datasets (Budzianowski et al., 2018; Byrne et al., 2019; Rastogi et al., 2019). Dialogue state tracking (DST) is a backbone of TOD systems, where it is responsible for extracting the user's goal represented as a set of slot-value pairs (e.g., (*area*, *center*), (*food*, *British*)), as illustrated in the upper part of Figure 1. The DST module's output is treated as the summary of the user's goal so far in the dialogue and directly consumed by the subsequent dialogue policy component to determine the system's next action and response. Hence, the accuracy of the DST module is critical to prevent downstream error propagation (Liu and Lane, 2018), affecting the end-to-end performance of the whole system.

With the advent of representation learning in NLP (Pennington et al., 2014; Devlin et al., 2019; Radford et al., 2019), the accuracy of DST models has increased from 15.8% (in 2018) to 55.7% (in 2020). While measuring the held-out accuracy is often useful, practitioners consistently overestimate their model's generalization (Ribeiro et al., 2020; Patel et al., 2008) since test data is usually collected in the same way as training data. In line with this hypothesis, Table 1 demonstrates that there is a substantial overlap of the slot values between training and evaluation sets of the MultiWOZ DST benchmark (Budzianowski et al., 2018). In Table 2, we observe that the slot co-occurrence distributions for evaluation sets tightly align with that of train split, hinting towards the potential

---

[*]Equal Contribution. Work was done during Shiyang's internship at Salesforce Research.

[1]Code is available at `https://github.com/salesforce/coco-dst`

| data | attraction-name | hotel-name | restaurant-name | taxi-departure | taxi-destination | train-departure | train-destination |
|------|-----------------|------------|-----------------|----------------|------------------|-----------------|-------------------|
| dev | 94.5 | 96.4 | 97.3 | 98.6 | 98.2 | 99.6 | 99.6 |
| test | 96.2 | 98.4 | 96.8 | 95.6 | 99.5 | 99.4 | 99.4 |

Table 1: The percentage (%) of domain-slot values in dev/test sets covered by training data.

| slot name | data | area | book day | book time | food | name | price range |
|-----------|------|------|----------|-----------|------|------|-------------|
| | train | 1.9 | 38.8 | 39.2 | 2.1 | 16.4 | 1.5 |
| book people | dev | 1.9 | 38.9 | 38.9 | 1.9 | 16.3 | 2.2 |
| | test | 2.7 | 36.9 | 37.7 | 1.6 | 18.7 | 2.4 |

Table 2: Co-occurrence distribution(%) of *book people* slot with other slots in *restaurant* domain within the same user utterance. It rarely co-occurs with particulars slots (e.g., *food*), which hinders the evaluation of DST models on realistic user utterances such as "*I want to book a Chinese restaurant for 8 people.*"

limitation of the held-out accuracy in reflecting the actual generalization capability of DST models. Inspired by this phenomenon, we aim to address and provide insights into the following question: *how well do state-of-the-art DST models generalize to the novel but realistic scenarios that are not captured well enough by the held-out evaluation set?*

Most prior work (Iyyer et al., 2018; Jin et al., 2019) focus on adversarial example generation for robustness evaluation. They often rely on perturbations made directly on test examples in the held-out set and assume direct access to evaluated models' gradients or outputs. Adversarial examples generated by these methods are often unnatural or obtained to hurt target models deliberately. It is imperative to emphasize here that both our primary goal and approach significantly differ from the previous line of work: (i) Our goal is to evaluate DST models beyond held-out accuracy, (ii) We leverage turn-level structured meaning representation (belief state) along with its dialogue history as conditions to generate user response without relying on the original user utterance, (iii) Our approach is entirely model-agnostic, assuming no access to evaluated DST models, (iv) Perhaps most importantly, we aim to produce novel but realistic and meaningful conversation scenarios rather than intentionally adversarial ones.

We propose *controllable counterfactuals* (CoCo) as a principled, model-agnostic approach to generate novel scenarios beyond the held-out conversations. Our approach is inspired by the combination of two natural questions: how would DST systems react to (1) unseen slot values and (2) rare but realistic slot combinations? CoCo first encapsulates these two aspects under a unified concept called *counterfactual goal* obtained by a stochastic policy of dropping and adding slots to the original turn-level belief state followed by replacing slot values. In the second step, CoCo conditions on the dialogue history and the counterfactual goal to generate *counterfactual conversation*. We cast the actual utterance generation as a conditional language modeling objective. This formulation allows us to plug-in a pretrained encoder-decoder architecture (Raffel et al., 2020) as the backbone that powers the counterfactual conversation generation. We also propose a strategy to filter utterances that fail to reflect the counterfactual goal exactly. We consider *value substitution* (VS), as presented in Figure 1, as a special CoCo case that only replaces the slot values in the original utterance without adding or dropping slots. When we use VS as a fall-back strategy for CoCo (i.e., apply VS when CoCo fails to generate valid user responses after filtering), we call it CoCo+.

Evaluating three strong DST models (Wu et al., 2019; Heck et al., 2020; Hosseini-Asl et al., 2020) with our proposed controllable counterfactuals generated by CoCo and CoCo+ shows that the performance of each significantly drops (up to 30.8%) compared to their joint goal accuracy on the original MultiWOZ held-out evaluation set. On the other hand, we find that these models are, in fact, quite robust to paraphrasing with back-translation, where their performance only drops up to 2%. Analyzing the effect of data augmentation with CoCo+ shows that it consistently improves the robustness of the investigated DST models on counterfactual conversations generated by each of VS, CoCo and CoCo+. More interestingly, the same data augmentation strategy improves the joint goal accuracy of the best of these strong DST models by 1.3% on the original MultiWOZ evaluation set. Human evaluations show that CoCo-generated counterfactual conversations perfectly reflect the underlying user goal with more than 95% accuracy and are found to be quite close to original conversations in terms of their human-like scoring. This further proves our proposed approach's reliability and potential to be adopted as part of DST models' robustness evaluation.

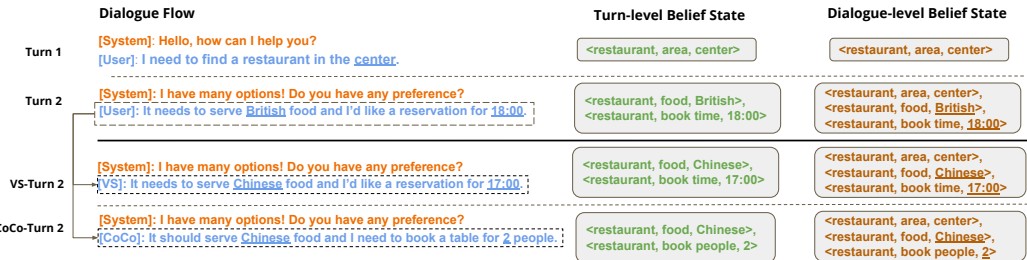

Figure 1: The upper left is a dialogue example between *user* and *system* with its turn-level and dialogue-level belief states on the upper right. The lower left are valid user utterance variations generated by VS and CoCo with their corresponding belief states derived from the original ones on the right.

## 2 RELATED WORK

**Dialogue State Tracking.** DST has been a core component in current state-of-the-art TOD systems. Traditional approaches usually rely on hand-crafted features or domain-specific lexicon (Henderson et al., 2014; Wen et al., 2017) and require a predefined ontology, making them hard to extend to unseen values. To tackle this issue, various methods have been proposed. Gao et al. (2019) treats DST as a reading comprehension problem and predicts slot values with start and end positions in the dialogue context. Zhang et al. (2019a) proposes DS-DST, a dual-strategy model that predicts values in domains with a few possible candidates from classifiers and others from span extractors. Furthermore, Heck et al. (2020) proposes TripPy, a triple copy strategy model, which allows it to copy values from the context, previous turns' predictions and system informs.

An alternative to classification and span prediction is value generation. Wu et al. (2019) generates slot values with a pointer generator network See et al. (2017) without relying on fixed vocabularies and spans. (Hosseini-Asl et al., 2020) models DST as a conditional generation problem and directly finetunes GPT2 (Radford et al., 2019) on DST task and achieves state-of-the-art on the MultiWOZ.

**Adversarial Example Generation.** Adversarial example generation has been commonly studied in computer vision (Szegedy et al., 2014; Goodfellow et al., 2015). Recently, it has received growing attention in NLP domain as well. Papernot et al. (2016) finds adversarial examples in the embedding space, and then remapped them to the discrete space. Alzantot et al. (2018) proposes a population-based word replacing method and aims to generate fluent adversarial sentences. These methods often edit the original data greedily assuming access to the model's gradients or outputs besides querying the underlying model many times (Jin et al., 2019). Alternative line of work investigates generating adversarial examples in a model-agnostic way. Iyyer et al. (2018) proposes to generate adversarial paraphrases of original data with different syntactic structures. Jia and Liang (2017) automatically generates sentences with key word overlappings of questions in SQuAD (Rajpurkar et al., 2016) to distract computer systems without changing the correct answer or misleading humans.

Although different methods have been proposed to evaluate the robustness of NLP models, majority of the prior work in this line focus either on text classification, neural machine translation or reading comprehension problems. Perhaps the most similar existing work with ours are (Einolghozati et al., 2019) and (Cheng et al., 2019). Einolghozati et al. (2019) focuses on intent classification and slot tagging in TOD while Cheng et al. (2019) targets at synthetic competitive negotiation dialogues (Lewis et al., 2017) without DST component. In this work, however, we focus on evaluating a core component of state-of-the-art TOD, DST, on the widely used benchmark, MultiWOZ. To the best of our knowledge, ours is the first work to systematically evaluate the robustness of DST models.

## 3 BACKGROUND

**Multi-domain DST task definition.** Let $X_t = \{(U_1^{\text{sys}}, U_1^{\text{usr}}), ..., (U_t^{\text{sys}}, U_t^{\text{usr}})\}$ denote a sequence of turns of a dialogue until the $t$-th turn, where $U_i^{\text{sys}}$ and $U_i^{\text{usr}}$ ($1 \leq i \leq t$) denote system and user utterance at the $i$-th turn, respectively. In multi-domain DST, each turn $(U_i^{\text{sys}}, U_i^{\text{usr}})$ talks about a specific domain (e.g., *hotel*), and a certain number of slots (e.g., *price range*) in that domain. We denote all $N$ possible domain-slot pairs as $S = \{S_1, ...S_N\}$. The task is to track the value for each

$S_j$ ($1 \leq j \leq N$) over $X_t$ (e.g., *hotel-price range*, *cheap*). Belief states can be considered at two granularities: turn-level ($L_t$) and dialog-level ($B_t$). $L_t$ tracks the information introduced in the last turn while $B_t$ tracks the accumulated state from the first turn to the last. As illustrated in the upper part of Figure 1, when the dialogue flow arrives at the second turn, $B_2$ becomes {(*restaurant-area*, *center*), (*restaurant-food*, *British*), (*restaurant-book time*, *18:00*)}, while $L_2$ is {(*restaurant-food*, *British*), (*restaurant-book time*, *18:00*)}, essentially tracking the update to $B_t$ by the last turn.

**Problem definition.** Given a tuple $< X_t, L_t, B_t >$, our goal is to generate a new user utterance $\hat{U}_t^{\text{usr}}$ to form a novel conversation scenario $\hat{X}_t = \{(U_1^{\text{sys}}, U_1^{\text{usr}}), ..., (U_t^{\text{sys}}, \hat{U}_t^{\text{usr}})\}$ by replacing the original user utterance $U_t^{\text{usr}}$ with $\hat{U}_t^{\text{usr}}$. To preserve the coherence of dialogue flow, we cast the problem as generating an alternative user utterance $\hat{U}_t^{\text{usr}}$ conditioned on a modified $\hat{L}_t$ derived from original turn-level belief state $L_t$ in a way that is consistent with the global belief state $B_t$. This formulation naturally allows for producing a new tuple $< \hat{X}_t, \hat{L}_t, \hat{B}_t >$ controllable by $\hat{L}_t$, where $\hat{B}_t$ is induced by $B_t$ based on the difference between $L_t$ and $\hat{L}_t$. As illustrated in the lower part of Figure 1, $U_2^{\text{usr}}$ is replaced with the two alternative utterances that are natural and coherent with the dialogue history. We propose to use the resulting set of $< \hat{X}_t, \hat{L}_t, \hat{B}_t >$ to probe the DST models.

**Paraphrase baseline with back-translation.** Paraphrasing the original utterance $U_t^{\text{usr}}$ is a natural way to generate $\hat{U}_t^{\text{usr}}$. With the availability of advanced neural machine translation (NMT) models, round-trip translation between two languages (i.e., back-translation (BT)) has become a widely used method to obtain paraphrases for downstream applications (Yu et al., 2018). We use publicly available pretrained *English→German* ($\log(g|e)$) and *German→English* ($\log(e|g)$) NMT models.[2] We translate $U_t^{\text{usr}}$ from English to German with a beam size $K$, and then translate each of the $K$ hypotheses back to English with the beam size $K$. Consequently, we generate $K^2$ paraphrase candidates of $\hat{U}_t^{\text{usr}}$ and then rank them according to their round-trip confidence score $\log(g|e) + \log(e|g)$. As paraphrases are expected to preserve the meaning of $U_t^{\text{usr}}$, we set $\hat{L}_t = L_t$ and $\hat{B}_t = B_t$.

## 4 CoCo

As illustrated in Figure 2, CoCo consists of three main pillars. We first train a conditional user utterance generation model $p_\theta(U_t^{\text{usr}}|U_t^{\text{sys}}, L_t)$ using original dialogues. Secondly, we modify $L_t$ into a possibly arbitrary $\hat{L}_t$ by our counterfactual goal generator. Given $\hat{L}_t$ and $U_t^{\text{sys}}$, we sample $\hat{U}_t^{\text{usr}} \sim p_\theta(\hat{U}_t^{\text{usr}}|U_t^{\text{sys}}, \hat{L}_t)$ with beam search followed by two orthogonal filtering mechanisms to further eliminate user utterances that fail to reflect the counterfactual goal $\hat{L}_t$.

### 4.1 VALUE SUBSTITUTION

A robust DST model should correctly reflect value changes in user utterances when tracking user's goal. However, slot-value combinations, e.g. (*restaurant-book time*, *18:00*), in evaluation sets are limited and even have significant overlaps with training data as shown in Table 1. To evaluate DST models with more diverse patterns, we propose a Value Substitution (VS) method to generate $\hat{U}_t^{\text{usr}}$. Specifically, for each value of $S_j$ in $L_t$, if the value only appears in $U_t^{\text{usr}}$ rather than $U_t^{\text{sys}}$, we allow it to be substituted. Otherwise, we keep it as is. This heuristic is based on the following three observations: (1) if the value comes from $U_t^{\text{sys}}$, e.g. TOD system's recommendation of restaurant food, changing it may make the dialogue flow less natural and coherent (2) if it never appears in the dialogue flow, e.g. *yes* of *hotel-parking*, changing it may cause belief state label errors (3) if it only appears in $U_t^{\text{usr}}$, it is expected that changing the value won't cause issues in (1) and (2).

For values that can be substituted, new values are sampled from a *Slot-Value Dictionary*, a predefined value set for each domain-slot. These new values are then used to update their counterparts in $U_t^{\text{usr}}$, $L_t$ and $B_t$. We defer the details of slot-value dictionary to section 4.2. After the update, we get $\hat{U}_t^{\text{usr}}$, $\hat{L}_t$ and $\hat{B}_t$, and can use $< \hat{X}_t, \hat{L}_t, \hat{B}_t >$ to evaluate the performance of DST models. An example of how VS works is illustrated in the lower part of Figure 1. At the second turn, as *British* and *18:00* are in $L_2$ and only appear in $U_2^{\text{usr}}$ rather than $U_2^{\text{sys}}$, we can replace them with *Chinese* and *17:00* that

---

[2] https://pytorch.org/hub/pytorch_fairseq_translation

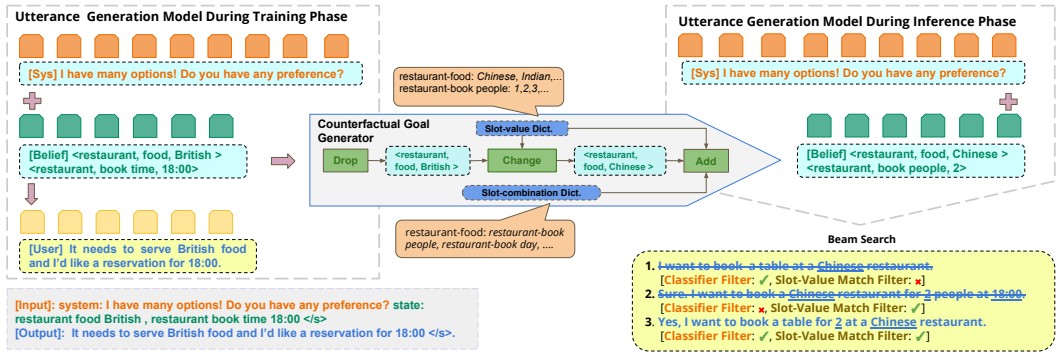

Figure 2: The overall pipeline of CoCo. The very left part represents the training phase of utterance generation model, where the concatenation of $U_t^{\text{sys}}$ and $L_t$ is processed by the encoder, which the decoder then conditions on to generate the user utterance $U_t^{\text{usr}}$. The input and output of this model is shown within the box at the lower-left. The right part depicts the inference phase, where the counterfactual goal generator first modifies the original belief $L_t$ fed from the left part into a new one $\hat{L}_t$, which is then fed to the trained utterance generator along with the same conversation history to generate $\hat{U}_t^{\text{usr}}$ by beam search followed by filtering undesired utterances. Note that conversational turns in inference phase don't have to originate from training phase.

are sampled from a slot-value dictionary, respectively, to get $\hat{U}_2^{\text{usr}}$, $\hat{L}_2$ and $\hat{X}_2$ without interrupting the naturalness of the dialogue flow.

## 4.2 CONTROLLABLE COUNTERFACTUAL GENERATION

Back-translation (BT) and value-substitution (VS) provide controllability at different granularities. BT only provides syntactic variety while preserving the meaning, hence the belief state. VS can only replace the values of the existing slots in an utterance while still having to exactly retain all the slots. However, neither of them are able to explore conversations with even slightly modified set of slots. We propose a principled approach to unlock the capability of conversation generation that generalizes beyond just transformation of existing utterances. We cast it as a task of generating novel user utterances ($U_t^{\text{usr}}$) from a given conversation history ($U_t^{\text{sys}}$) and a turn-level user goal ($L_t$).

We propose to tackle this problem with a conditional generation model that utilizes a pre-trained encoder-decoder architecture (Raffel et al., 2020; Lewis et al., 2020) to approximate $p(U_t^{\text{usr}}|U_t^{\text{sys}}, L_t)$, where the concatenation of $U_t^{\text{sys}}$ and $L_t$ is used as input to the encoder and $U_t^{\text{usr}}$ is set to be the target sequence to be generated by the decoder, as illustrated in the lower-left of Figure 2. To learn this distribution, we factorize it by chain rule (Bengio et al., 2003) and train a neural network with parameters $\theta$ to minimize the aggregated negative log-likelihood $\mathcal{J}_{\text{gen}}$ over each dialogue turn tuple $(U_t^{\text{sys}}, L_t, U_t^{\text{usr}})$ where $U_t^{\text{usr}} = (U_{t,1}^{\text{usr}}, U_{t,2}^{\text{usr}}, \ldots, U_{t,n_t}^{\text{usr}})$ and $U_{t,k}^{\text{usr}}$ is its $k$-th token:[3]

$$p_\theta(U_t^{\text{usr}}|U_t^{\text{sys}}, L_t) = \prod_{k=1}^{n_t} p_\theta(U_{t,k}^{\text{usr}}|U_{t,<k}^{\text{usr}}, U_t^{\text{sys}}, L_t), \quad \mathcal{J}_{\text{gen}} = -\sum_{k=1}^{n_t} \log p_\theta(U_{t,k}^{\text{usr}}|U_{t,<k}^{\text{usr}}, U_t^{\text{sys}}, L_t) \quad (1)$$

Once the parameters $\theta$ of the goal-conditioned utterance generation model $p_\theta$ are learned from these tuples, it gives us the unique ability to generate novel conversation turns by plugging in an arbitrary but consistent counterfactual goal $\hat{L}_t$ derived from $L_t$. An example of how the counterfactual goal generator operates is shown in the middle part of Figure 2. The counterfactual goal generator has three components, namely operation, slot-value dictionary and slot-combination dictionary.

**Operation** decides to apply which combination of the following three meta-operations, namely *drop*, *change* and *add* on $L_t$. *Drop* is used to remove values from a non-empty slot in $L_t$. *Change* borrows the same operation in VS, to substitute existing values. *Add* allows us to add new domain-slot values into $L_t$, giving us the power of generating valid but more complicated $\hat{U}_t^{\text{usr}}$.

---

[3]More details can be found in Appendix E.1.

**Slot-Value Dictionary** has a pre-defined value set $S_j^{\text{val}}$ for each $S_j$. Once *change* and/or *add* meta-operation is activated for $S_j$, counterfactual goal generator will randomly sample a value from $S_j^{\text{val}}$.

**Slot-Combination Dictionary** has a predefined domain-slot set $S_j^{\text{add}}$ for each $S_j$. When *add* meta-operation is activated, counterfactual goal generator will sample a domain-slot from the intersection among all $S_j^{add}$, where $S_j$ has non-empty values within $L_t$. Once a new domains-slot is sampled, its value will then be sampled from its corresponding value set as defined in slot-value dictionary.

Given $L_t$, the counterfactual goal generator first takes $L_t$ as its input, and sequentially applies *drop*, *change* and *add* to output $\hat{L}_t$. Given $\hat{L}_t$ and $U_t^{\text{sys}}$, we can sample $\hat{U}_t^{\text{usr}} \sim p_\theta(\hat{U}_t^{\text{usr}}|U_t^{\text{sys}}, \hat{L}_t)$ with beam search. We use a rule-based method to get $\hat{B}_t$ of $\hat{X}_t$. Specifically, we obtain $\bar{B}_{t-1}$ by calculating the set difference of $B_t$ and $L_t$. Given $\bar{B}_{t-1}$ and $\hat{L}_t$, we update the domain-slot in $\bar{B}_{t-1}$ if its value in $\hat{L}_t$ is not *none*, otherwise we keep its value as it is in $\bar{B}_{t-1}$ following (Chao and Lane, 2019). After the update, we get $\hat{B}_t$ and use it as the dialogue-level label of $\hat{X}_t$.

## 4.3 FILTERING

We have presented methods to generate $\hat{U}_t^{\text{usr}}$, but how do we make sure that the generated utterance correctly reflects the user goal represented by $\hat{L}_t$? To motivate our methods, we take an example generated by beam search located at the lower right of Figure 2 for illustration. In this example, the first hypothesis doesn't include value *2* for *restaurant-book people* that is within $\hat{L}_t$. On the contrary, the second hypothesis includes value *18:00* for *restaurant-book time* that is not part of $\hat{L}_t$. We call these two phenomenons *de-generation* and *over-generation*, respectively. Filtering candidates with these issues is thus an important step to make sure $(U_t^{\text{sys}}, \hat{U}_t^{\text{usr}})$ perfectly expresses the user goals in $\hat{L}_t$. We propose two filtering methods, namely *slot-value match filter* and *classifier filter*, to alleviate *de-generation* and *over-generation* issues, respectively.

**Slot-Value Match Filter.** To tackle with *de-generation* issue, we choose a subset of values in $\hat{L}_t$ (values that should only appear in $\hat{U}_t^{\text{usr}}$ rather than $U_t^{\text{sys}}$) to eliminate candidates that fail to contain all the values in the subset.[4] In Figure 2, the first hypothesis from the beam search output will be eliminated by this filter because it does not include the value *2* for *restaurant-book people* in $\hat{L}_t$.

**Classifier Filter.** As shown in Table 2, the slot *restaurant-book people* frequently appears together with *restaurant-book time* in the data used to train our generation model $p_\theta(\hat{U}_t^{\text{usr}}|U_t^{\text{sys}}, \hat{L}_t)$, which may cause the resulting generation model to fall into *over-generation* issue. To deal with this *over-generation* problem, we propose to use a N-way multi-label classifier to eliminate such candidates. We employ BERT-base (Devlin et al., 2019) as its backbone:

$$H_t^{\text{CLS}} = \text{BERT}([\text{CLS}] \oplus [X_{t-1}] \oplus [\text{SEP}] \oplus [U_t^{\text{sys}}] \oplus [U_t^{\text{usr}}]) \in \mathbb{R}^{d_{\text{emb}}} \tag{2}$$

where $H_t^{\text{CLS}} \in \mathbb{R}^{d_{\text{emb}}}$ is the representations of CLS token of BERT with dimension $d_{\text{emb}}$. We then feed $H_t^{\text{CLS}}$ into a linear projection layer followed by Sigmoid function:

$$P = \text{Sigmoid}(W(H_t^{\text{CLS}})) \in \mathbb{R}^N, \quad \mathcal{J}_{\text{cls}} = -\frac{1}{N}\sum_{j=1}^{N}(Y_j \cdot \log P_j + (1 - Y_j) \cdot \log(1 - P_j)) \tag{3}$$

where $W \in \mathbb{R}^{N \times d_{\text{emb}}}$ is the trainable weight of the linear projection layer and $P_j$ is probability that slot $S_j$ appears at $t$-th turn of $X_t$ with $Y_j$ as its label. The classifier is trained with $\mathcal{J}_{\text{cls}}$, i.e. the mean binary cross entropy loss of every slot $S_j$ and achieves a precision of 92.3% and a recall of 93.5% on the development set [5]. During inference, the classifier takes $\hat{X}_t$ as input and predicts whether a slot $S_i$ appears at $t$-th turn or not with threshold 0.5. We use this filter to eliminate generated candidates for which the classifier predicts at least one slot $S_j$ mentioned in $(U_t^{\text{sys}}, \hat{U}_t^{\text{usr}})$ while $S_j \notin \hat{L}_t$. In Figure 2, our classifier filter eliminates the second hypothesis from the output of beam search because $\hat{L}_t$ does not contain the slot *restaurant-book time* while it is mentioned in the generated utterance.

---

[4]For *hotel-parking* and *hotel-internet*, we use *parking* and *wifi* as their corresponding values for filtering.

[5]We defer further details of the classifier to Appendix E.2.

## 5 EXPERIMENTS

### 5.1 EXPERIMENTAL SETUP

We consider three strong multi-domain DST models to evaluate the effect of COCO-generated counterfactual conversations in several scenarios. TRADE (Wu et al., 2019) builds upon pointer generator network and contains a slot classification gate and a state generator module to generate states. TRIPPY (Heck et al., 2020) introduces a classification gate and a triple copy module, allowing the model to copy values either from the conversation context or previous turns' predictions or system informs. SIMPLETOD (Hosseini-Asl et al., 2020) models DST as a conditional generation problem with conversation history as its condition and belief state as its target and finetunes on GPT2.

**Evaluation.** We train each of these three models following their publicly released implementations on the standard train/dev/test split of MultiWOZ 2.1 (Eric et al., 2019). We use the joint goal accuracy to evaluate the performance of DST models. It is 1.0 if and only if the set of (*domain-slot*, *value*) pairs in the model output exactly matches the oracle one, otherwise 0.

**Slot-Value Dictionary.** We carefully design two sets of slot-value dictionaries to capture the effect of unseen slot values from two perspectives, namely *in-domain* (*I*) and *out-of-domain* (*O*). *I* is a dictionary that maps each slot to a set of values that appear in MultiWOZ test set, but not in the training set.[6] On the other hand, we construct *O* using external values (e.g., hotel names from Wikipedia) that fall completely outside of the MultiWOZ data for the slots (e.g., *hotel-name*, *restaurant-name*, etc.). Otherwise, we follow a similar fall-back strategy for slots (e.g., *hotel-internet*) with no possible external values beyond the ones (e.g., *yes* and *no*) in the original data.

**Slot-Combination Dictionary.** As illustrated in Table 2, held-out evaluation set follows almost the same slot co-occurrence distribution with training data. This makes it difficult to estimate how well DST models would generalize on the valid conversation scenarios that just do not obey the same distribution. COCO's flexibility at generating a conversation for an arbitrary turn-level belief state naturally allows us to seek an answer to this question. To this end, we design three slot combination dictionaries, namely *freq*, *neu* and *rare*. A slot combination dictionary directly controls how different slots can be combined while generating counterfactual goals. As suggested by their names, *freq* contains frequently co-occurring slot combinations (e.g., *book people* is combined only with *book day* and *book time* slots), while *rare* is the opposite of *freq* grouping rarely co-occurring slots together, and *neu* is more neutral allowing any meaningful combination within the same domain.[7]

### 5.2 MAIN RESULTS

Before reporting our results, it is important to note that several different post-processing strategies are used by different DST models. To make a fair comparison across different models, we follow the same post-processing strategy employed by SIMPLETOD evaluation script for TRADE and TRIPPY as well. We summarize our main results in Figure 3. While all three DST models are quite robust to back-translation (BT)[8], their performance significantly drop on counterfactual conversations generated by each of VS, COCO and COCO+ compared to MultiWOZ held-out set accuracy (original).

**Unseen Slot-Value Generalization.** We analyze the effect of unseen slot values for the two dictionaries (*I* and *O*) introduced in the previous section compared to the original set of slot values that have large overlap with the training data. Results presented on the left part of Figure 3 show that the performance of DST models significantly drops up to 11.8% compared to original accuracy even on the simple counterfactuals generated by VS strategy using in-domain unseen slot-value dictionary (I). Furthermore, using out-of-domain slot-value dictionary (O) results in about 10% additional drop in accuracy consistently across the three models. Consistent and similar drop in accuracy suggests that TRADE, SIMPLETOD, and TRIPPY are almost equally susceptible to unseen slot values.

**Generalization to Novel Scenarios.** The right section of Figure 3 presents the main results in our effort to answer the central question we posed at the beginning of this paper. Based on these re-

---

[6]When this set is empty for a slot (e.g., *hotel-area*), we use the set of all possible values (e.g., *center*, *east*, *west*, *south*, *north*) for this slot from training data. Please see Appendix I for further details.

[7]Please see Appendix H for further details.

[8]Similar to COCO, we back-translate only the turns with non-empty turn-level belief states and apply slot-value match filter. We fall back to original user utterance if none of the paraphrases passes the filter.

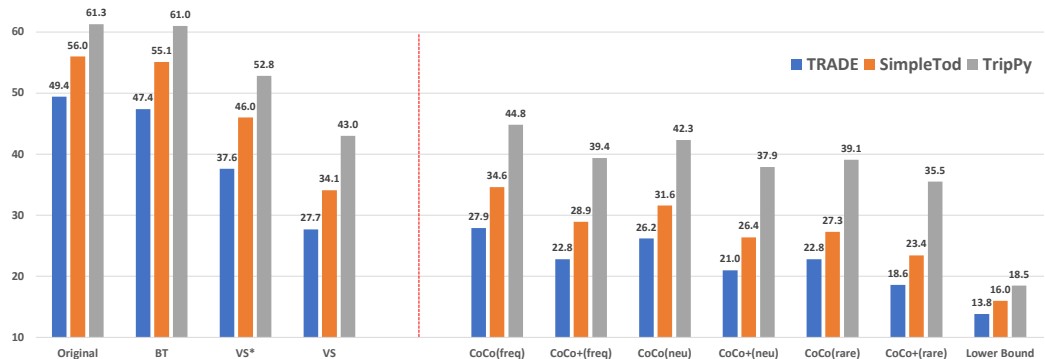

Figure 3: Joint goal accuracy (%) across different methods. "Original" refers to the results on the original held-out test set. * denotes results obtained from in-domain unseen slot-value dictionary (*I*). VS, CoCo and CoCo+ results use out-of-domain slot-value dictionary (*O*). For brevity, we omit CoCo and CoCo+ results using in-domain slot-value dictionary. See Appendix C for the full results. *freq*, *neu*, and *rare* indicate which slot-combination dictionary is used. Lower bound refers to the percentage of correct predictions on turns with empty turn-level belief state over original held-out test set.

sults, we see that state-of-the-art DST models are having a serious difficulty generalizing to novel scenarios generated by both CoCo and CoCo+ using three different slot combination strategies. The generalization difficulty becomes even more serious on counterfactuals generated by CoCo+. As expected, the performance drop consistently increases as we start combining less and less frequently co-occurring slots (ranging from *freq* to *rare*) while generating our counterfactual goals. In particular, CoCo+(rare) counterfactuals drops the accuracy of TRADE from 49.4% to 18.6%, pushing its performance very close to its lower bound of 13.8%. Even the performance of the most robust model (TRIPPY) among the three drops by up to 25.8%, concluding that held-out accuracy for state-of-the-art DST models may not sufficiently reflect their generalization capabilities.

**Transferability Across Models.** As highlighted before, a significant difference and advantage of our proposed approach lies in its model-agnostic nature, making it immediately applicable for evaluation of any DST model. As can be inferred from Figure 3, the effect of CoCo-generated counterfactuals on the joint goal accuracy is quite consistent across all three DST models. This result empirically proves the transferability of CoCo, strengthening its reliability and applicability to be generally employed as a robustness evaluation of DST models by the future research.

## 5.3 HUMAN EVALUATION

We next examine the quality of our generated data from two perspectives: "human likeliness" and "turn-level belief state correctness". The human likeliness evaluates whether a user utterance is fluent and consistent with its dialog context. The turn-level belief state correctness evaluates whether $(U_t^{\text{sys}}, \hat{U}_t^{\text{usr}})$ exactly expresses goals in $\hat{L}_t$. Both metrics are based on binary evaluation. We randomly sample 100 turns in the original test data and their corresponding CoCo-generated ones.

|  | Human likeliness | Correctness |
|---|---|---|
| Human | 87% | 85% |
| CoCo(ori) | 90% | 91% |
| CoCo(freq) | 90% | 99% |
| CoCo(neu) | 79% | 98% |
| CoCo(rare) | 82% | 96% |

Table 3: Human evaluation.

For the CoCo-generated data, we have two different settings to examine its quality. The first is to use the original turn-level belief state to generate user utterance, denoted by CoCo(ori). The second setting is to verify the quality of the conversations generated by CoCo(freq)-, CoCo(neu)- and CoCo(rare) as they hurt the DST models' accuracy significantly as shown in Figure 3. For each result row reported in Table 3, we ask three individuals with proficient English and advanced NLP background to conduct the evaluation, and use majority voting to determine the final scores.

We can see that CoCo(ori) generated conversations are almost as human-like as original conversations. Furthermore, CoCo(ori) generated slightly more "correct" responses than the original utterances in MultiWoZ 2.1. A presumable reason is that annotation errors exist in MultiWoZ 2.1, while our CoCo are trained on recently released cleaner MultiWoZ 2.2, making generated data have higher quality. In addition, all three variants of the CoCo-generated conversations consistently outper-

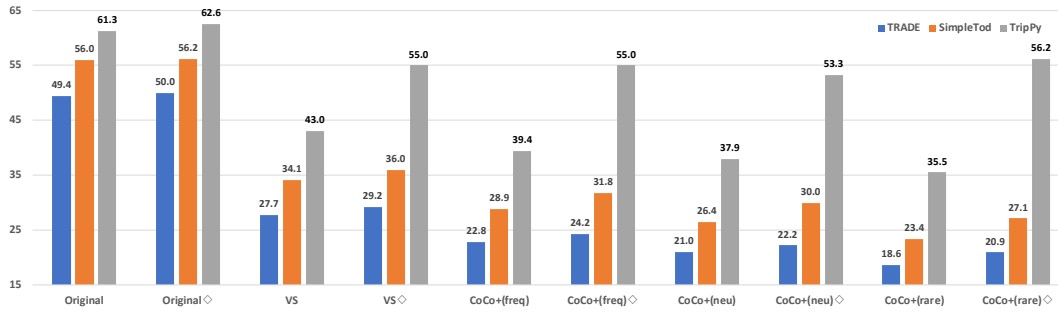

Figure 4: Comparison of retrained DST models (indicated by ◇ ) on CoCo+(rare)-augmented training data with their counterparts trained on original MultiWOZ train split.

form human response in terms of the turn-level belief state correctness. Although CoCo(neu) and CoCo(rare) are slightly less human-like than the original human response, CoCo(freq)-generated utterances have similar human-likeness as original ones. These results demonstrate the effectiveness of our proposed approach in generating not only high-fidelity but also human-like user utterances, proving its potential to be adopted as part of robustness evaluation of DST models.

## 5.4 ANALYSIS OF CoCo+ AS DATA AUGMENTATION DEFENSE

So far, we have focused on the generalization capability of DST models on CoCo-generated conversations using different slot-value and slot-combination dictionaries. We have observed that all three DST models are consistently most susceptible to conversations generated by CoCo+(rare) strategy. Instead, we now seek to answer the following question: *Would using conversations generated by CoCo+(rare) to augment the training data help these DST models in better generalizing to unseen slot values and/or novel scenarios?* Towards exploring this direction in a principled way, we design a new slot value dictionary (*train-O*) similar to out-of-domain unseen slot-value dictionary (*O*). For a fair comparison, we make sure that the slot values in *train-O* (please refer to Appendix I for the complete dictionary) do not overlap with the one (*O*) used for generating test conversations.

We first retrain each DST model on the MultiWOZ training split augmented with CoCo+(rare)-generated conversations using *train-O* slot-value dictionary. Retrained DST models are then evaluated on original test set as well as on the counterfactual test sets generated by VS and various versions of CoCo+. Results presented in Figure 4 show that retraining on the CoCo+(rare)-augmented training data improves the robustness of all three DST models across the board. Most notably, it rebounds the performance of TRIPPY on CoCo+(rare)-generated test set from 35.5% to 56.2%, significantly closing the gap with its performance (61.3%) on the original test set. We also observe that retrained DST models obtain an improved joint goal accuracy on the original MultiWOZ test set compared to their counterparts trained only on the original MultiWOZ train split, further validating the quality of CoCo-generated conversations. Finally, we would like to highlight that retrained TRIPPY achieves 62.6% joint goal accuracy, improving the previous state-of-the-art by 1.3%. We leave the exploration of how to fully harness CoCo as a data augmentation approach as future work.

## 6 CONCLUSION

We propose a principled, model-agnostic approach (CoCo) to evaluate dialogue state trackers beyond the held-out evaluation set. We show that state-of-the-art DST models' performance significantly drop when evaluated on the CoCo-generated conversations. Human evaluations validate that they have high-fidelity and are human-like. Hence, we conclude that these strong DST models have difficulty in generalizing to novel scenarios with unseen slot values and rare slot combinations, confirming the limitations of relying only on the held-out accuracy. When explored as a data augmentation method, CoCo consistently improves state-of-the-art DST models not only on the CoCo-generated evaluation set but also on the original test set. This further proves the benefit and potential of our approach to be adopted as part of a more comprehensive evaluation of DST models.

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

APPENDIX

## A  SLOT-LEVEL ANALYSIS

**Closer Look at the Effect of COCO+(rare) on TRIPPY.** In Figure 5, we take a closer look at the robustness of TRIPPY through slot-level analysis across three major scenarios. Comparison of *blue* and *orange* lines reveals that counterfactuals generated by COCO+(rare) consistently drops the performance of TRIPPY model (trained on the original MultiWOZ train split) across all the slots, significantly hurting the accuracy of most slots in *train* domain along with *book day* slot for *hotel* domain. On the other hand, comparing *green* and *orange* lines clearly demonstrates the effectiveness of COCO+(rare) as a data augmentation defense (see Section 5.4 for further details), assisting TRIPPY in recovering from most of the errors it made on COCO+(rare) evaluation set. In fact, it rebounds the joint goal accuracy of TRIPPY from 35.5% to 56.2% as presented more quantitatively in Figure 4.

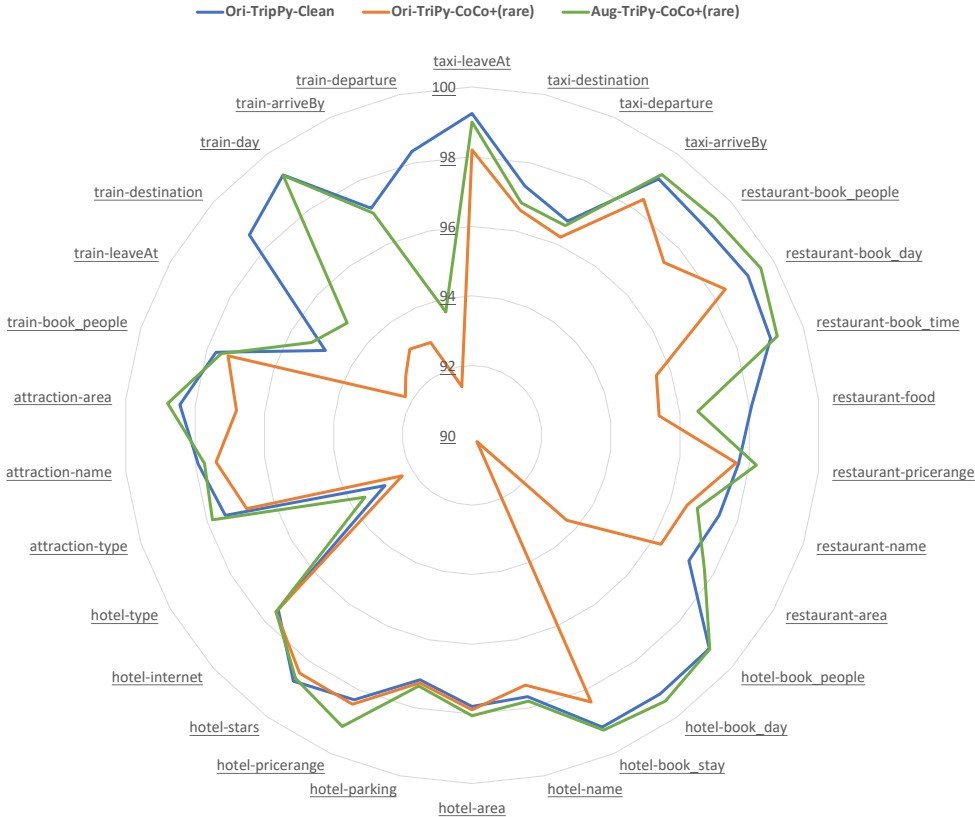

Figure 5: Slot-level accuracy analysis of TRIPPY. "Ori-TripPy-Clean" (blue) and "Ori-TripPy-CoCo+(rare)" (orange) denote TRIPPY (trained on original MultiWOZ training data) when evaluated against original test set and COCO+(rare) generated test set, respectively. "Aug-TripPy-CoCo+(rare)" (green) indicates slot-level accuracy of TRIPPY after data augmentation (see Section 5.4 for further details) when evaluated against test set generated by COCO+(rare).

## B    ABLATION STUDY ON OPERATIONS

In Table 4, we present ablation results on three meta operations (i.e., *drop*, *change*, *add*) that are used to generate counterfactual goals. The result in the first row corresponds to the performance of three DST models on evaluation set generated by CoCo including all three meta operations along with the classifier filter. Each row analyzes the effects of the corresponding meta operation or classifier by removing it from full models. From Table 4, we observe that removing *drop* operation from full models hurts the performance of the three models further. This may indicate that the investigated DST models are more vulnerable against user utterances including more slot combinations.

| CoCo | TRADE | SIMPLETOD | TRIPPY |
|---|---|---|---|
| Full | 26.2 | 31.6 | 42.3 |
| -Drop | 25.7 | 31.1 | 42.1 |
| -Add | 30.4 | 36.0 | 50.4 |
| -Change | 34.1 | 40.9 | 48.3 |
| -Classifier | 25.3 | 30.5 | 41.3 |

Table 4: Ablation study on the meta operations and classifier based filtering.

## C    FULL FIGURE FOR MAIN RESULT

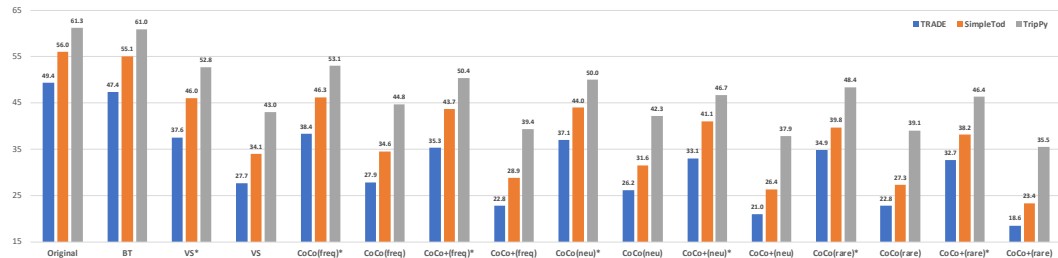

Figure 6: Joint goal accuracy (%) across different methods. "Original" refers to the results on the original held-out test set. * denotes results obtained from in-domain unseen slot-value dictionary ($I$) while other results use out-of-domain slot-value dictionary ($O$). *freq*, *neu*, and *rare* indicate which slot-combination dictionary is used.

## D    COCO+ MULTI-ROUND DATA AUGMENTATION ON TRIPPY

Section 5.4 shows that CoCo+ as data augmentation (COCOAUG) improves TRIPPY's joint goal accuracy by 1.3% when evaluated on the original test set following the post-processing strategy employed by SIMPLETOD. In this section, we further extend previous single-round data augmentation into multiple rounds. Specifically, for each tuple $< X_t, L_t, B_t >$ in the original training set, we can generate multiple $< \hat{X}_t, \hat{L}_t, \hat{B}_t >$ by sampling $\hat{L}_t$ multiple times and utilizing CoCo+ to generate corresponding $\hat{X}_t$ and $\hat{B}_t$. With this approach, generated multiple $< \hat{X}_t, \hat{L}_t, \hat{B}_t >$ combined with original $< X_t, L_t, B_t >$ can be used to train DST models.

We experiment with $\{1, 2, 4, 8\}$ times data augmentation size over original training data on TRIPPY following its own default cleaning so that results with previous methods are comparable. Comparison results with different baselines and data augmentation sizes are summarized in Table 5. When using more and more CoCo+ generated training data, TRIPPY gains benefits from more training data and consistently improves over baselines. When using 8x CoCo+ generated training data, TRIPPY provides 5.49% improvement over its counterpart without data augmentation. Furthermore, it achieves the new state-of-the-art join goal accuracy[9], outperforming CONVBERT-DG+MULTI, which uses open-domain dialogues and DialoGLUE (Mehri et al., 2020) as additional training data.

---

[9]Code is available at `https://github.com/salesforce/coco-dst/tree/multi_fold_coco_aug`

| Model | JOINT GOAL ACCURACY |
|---|---|
| DSTreader (Gao et al., 2019) | 36.40%† |
| TRADE (Wu et al., 2019) | 45.60% † |
| MA-DST (Kumar et al., 2020) | 51.04% † |
| NA-DST (Le et al., 2020) | 49.04% † |
| DST-picklist (Zhang et al., 2019a) | 53.30% † |
| SST (Chen et al., 2020) | 55.23% † |
| MinTL(T5-small) (Lin et al., 2020) | 50.95% § |
| SimpleTOD (Hosseini-Asl et al., 2020) | 55.76% § |
| ConvBERT-DG+Multi (Mehri et al., 2020) | 58.70% §¶ |
| TRIPPY (Heck et al., 2020) | 55.04%* |
| + COCOAUG (1×) | 56.00% |
| + COCOAUG (2×) | 56.94% |
| + COCOAUG (4×) | 59.73% |
| + COCOAUG (8×) | **60.53**% |

Table 5: Joint goal accuracy results on MultiWOZ 2.1 (Eric et al., 2019) of different methods. The upper part are results of various baselines and lower part are results of TRIPPY without or with $\{1, 2, 4, 8\}$ times data augmentation size over original training data. †: results reported from (Zhang et al., 2019a). §: results reported in their original papers. *: results of our run based on their officially released code. ¶: results need open-domain dialogues and DialoGLUE data.

# E    MODEL DETAILS

## E.1    THE DETAILS OF CONTROLLABLE GENERATION MODEL

We instantiate $p_\theta(U_t^{\mathrm{usr}}|U_t^{\mathrm{sys}}, L_t)$ with `T5-small` (Raffel et al., 2020) and utilize MultiWOZ 2.2 as its training data since it's cleaner than previous versions (Zang et al., 2020). During training, we use Adam optimizer (Kingma and Ba, 2015) with an initial learning rate $5e-5$ and set linear warmup to be 200 steps. The batch size is set to 36 and training epoch is set to be 10. The maximum sequence length of both encoder and decoder is set to be 100. We select the best checkpoint according to lowest perplexity on development set.

## E.2    THE DETAILS OF CLASSIFIER FILTER

We employ `BERT-base-uncased` as the backbone of our classifier filter and train classifier filter with Adam optimizer (Kingma and Ba, 2015) on MultiWOZ 2.2 since it's cleaner than previous versions (Zang et al., 2020). We select the best checkpoint based on the highest recall on development set during training process. The best checkpoint achieves a precision of 92.3% and a recall of 93.5% on the development set of MultiWOZ 2.2 and, a precision of 93.1% and a recall of 91.6% on its original test set.

# F  DIVERSITY EVALUATION

| slot name | data | area | book day | book time | food | name | price range | entropy |
|---|---|---|---|---|---|---|---|---|
| book people | Ori-test | 2.7 | 36.9 | 37.7 | 1.6 | 18.7 | 2.4 | 0.57 |
| | CoCo-test | 3.6 | 38.5 | 25.2 | 15.6 | 14.8 | 2.2 | 0.65 |

Table 6: Original test set (*Ori-test*) and CoCo generated test set (*CoCo-test*) co-occurrence distribution(%) comparisons of *book people* slot with other slots in *restaurant* domain within the same user utterance. The distribution entropy of *CoCo-test* is higher than its counterpart of *Ori-test* with an upper bound 0.78 corresponding to uniform distribution, meaning that *CoCo-test* is more diverse compared to *Ori-test* in terms of slot combinations.

| Data | Distinct-1 ↑ | Distinct-2 ↑ | Distinct-3 ↑ | Distinct-4 ↑ |
|---|---|---|---|---|
| Ori-test | 0.009 | 0.051 | 0.105 | 0.151 |
| CoCo-test | 0.009 | 0.053 | 0.113 | 0.166 |

Table 7: Language diversity comparisons of data points between *Ori-test* and *CoCo-test*. We use unique n-gram ratio (Li et al., 2016) as our diversity metric. ↑ represents a higher number means more diversity. Overall, *CoCo-test* has similar (if not better) diversity scores compared to *Ori-test*.

## G  GENERATED EXAMPLES BY COCO

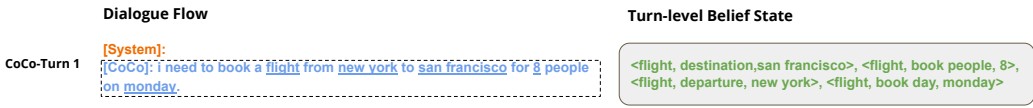

Figure 7: Zero-shot generation ability of CoCo on *flight* domain, which is never seen during training.

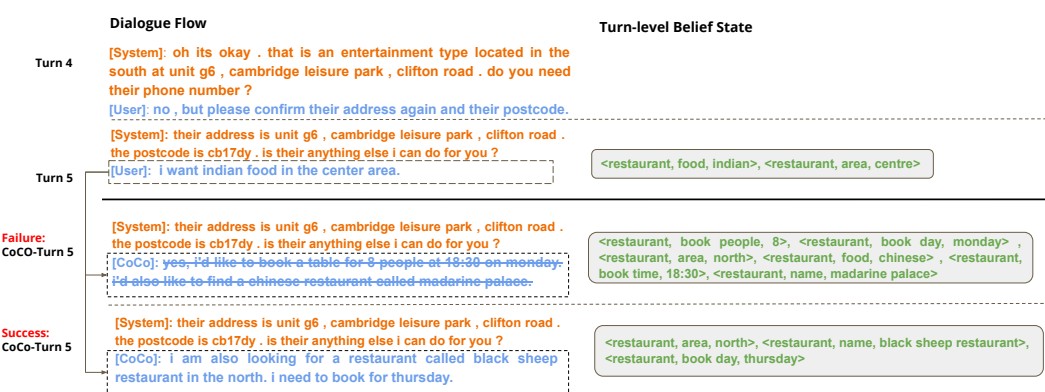

Figure 8: A success and failure example generated by CoCo with different slot-value combinations.

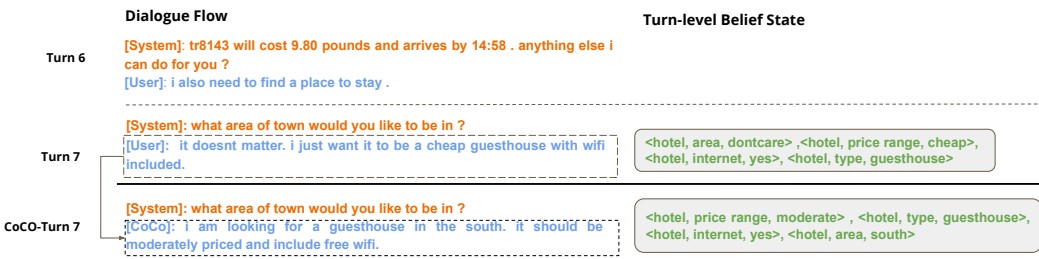

Figure 9: An example generated by CoCo with correct predictions by TRADE, SIMPLETOD and TRIPPY without retraining.

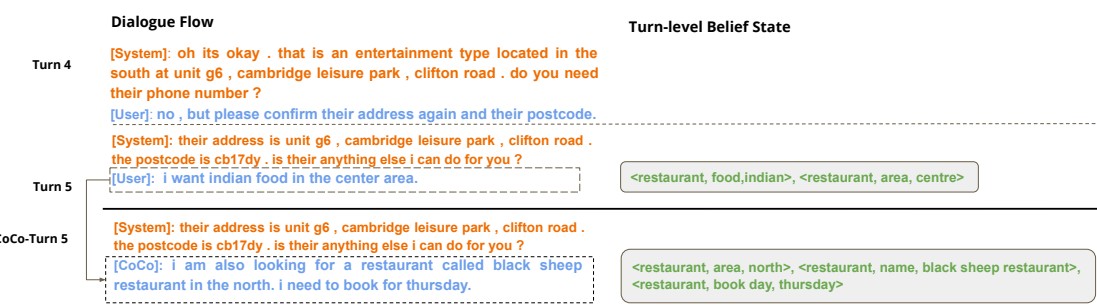

Figure 10: An example generated by CoCo with incorrect predictions by TRADE, SIMPLETOD and TRIPPY without retraining.

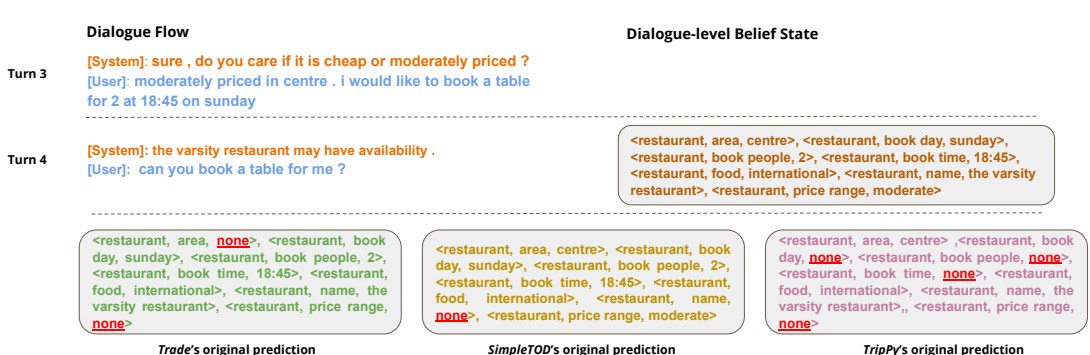

Figure 11: An example from original MultiWOZ test set, which is predicted incorrectly by original TRADE, SIMPLETOD and TRIPPY, is corrected by their retraining counterparts.

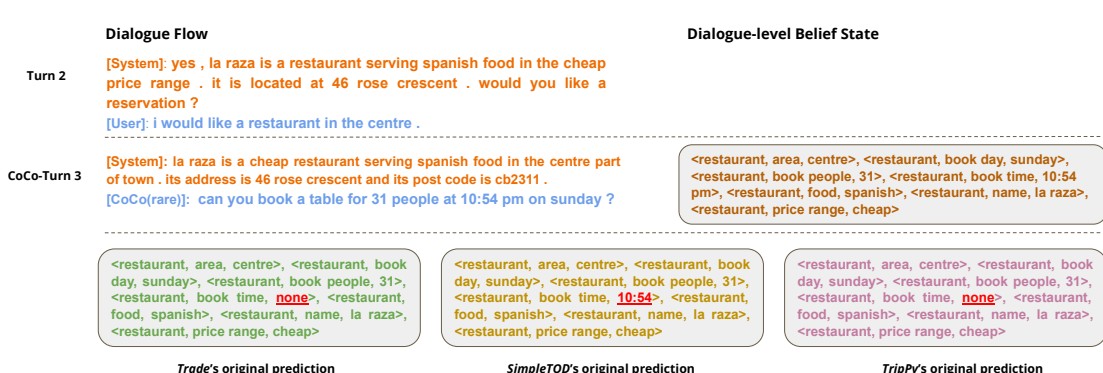

Figure 12: An example generated by CoCo(rare) evaluation set, which is predicted incorrectly by original TRADE, SIMPLETOD and TRIPPY, is corrected by their retraining counterparts.

# H SLOT-COMBINATION DICTIONARY

Please find the different slot-combination dictionaries introduced in the main paper below.

| domain-slot | *freq* |
|---|---|
| "hotel-internet" | ["hotel-area","hotel-parking","hotel-pricerange","hotel-stars","hotel-type"] |
| "hotel-type" | ["hotel-area","hotel-internet","hotel-parking","hotel-pricerange","hotel-stars"] |
| "hotel-parking" | ["hotel-area","hotel-internet","hotel-pricerange","hotel-stars","hotel-type"] |
| "hotel-pricerange" | ["hotel-area","hotel-internet","hotel-parking","hotel-stars","hotel-type"] |
| "hotel-book day" | ["hotel-book people","hotel-book stay"] |
| "hotel-book people": | ["hotel-book day","hotel-book stay"] |
| "hotel-book stay" | ["hotel-book day","hotel-book people"] |
| "hotel-stars" | ["hotel-area","hotel-internet","hotel-parking","hotel-pricerange","hotel-type"] |
| "hotel-area" | ["hotel-internet","hotel-parking","hotel-pricerange","hotel-stars","hotel-type"] |
| "hotel-name" | ["hotel-book day","hotel-book people","hotel-book stay"] |
| "restaurant-area" | ["restaurant-food","restaurant-pricerange"] |
| "restaurant-food" | ["restaurant-area","restaurant-pricerange"] |
| "restaurant-pricerange" | ["restaurant-area","restaurant-food"] |
| "restaurant-name" | ["restaurant-book day","restaurant-book people","restaurant-book time"] |
| "restaurant-book day" | ["restaurant-book people","restaurant-book time"] |
| "restaurant-book people" | ["restaurant-book day","restaurant-book time"] |
| "restaurant-book time" | ["restaurant-book day","restaurant-book people"] |
| "taxi-arriveby" | ["taxi-leaveat","train-book people"] |
| "taxi-leaveat" | ["taxi-arriveby","train-book people"] |
| "taxi-departure" | ["taxi-destination","taxi-leaveat","taxi-arriveby"] |
| "taxi-destination" | ["taxi-departure","taxi-arriveby","taxi-leaveat"] |
| "train-arriveby" | ["train-day","train-leaveat","train-book people"] |
| "train-departure" | ["train-arriveby","train-leaveat","train-destination","train-day","train-book people"] |
| "train-destination" | ["train-arriveby","train-leaveat","train-departure","train-day","train-book people"] |
| "train-day" | ["train-arriveby","train-leaveat","train-book people"] |
| "train-leaveat" | ["train-day"] |
| "train-book people" | [] |
| "attraction-name" | [] |
| "attraction-area" | ["attraction-type"] |
| "attraction-type" | ["attraction-area"] |

Table 8: Slot-combination dictionary for *freq* case.

| slot-name | *neu* |
|---|---|
| 'hotel-internet' | ['hotel-book day','hotel-name','hotel-book stay','hotel-pricerange', 'hotel-stars','hotel-area','hotel-book people','hotel-type','hotel-parking'] |
| 'hotel-area' | ['hotel-book day','hotel-name','hotel-book stay','hotel-pricerange', 'hotel-stars','hotel-book people','hotel-internet','hotel-type','hotel-parking'] |
| 'hotel-parking' | ['hotel-book day','hotel-name','hotel-book stay','hotel-pricerange','hotel-stars', 'hotel-area','hotel-book people','hotel-internet','hotel-type'] |
| 'hotel-pricerange' | ['hotel-book day','hotel-name','hotel-book stay','hotel-stars','hotel-area', 'hotel-book people','hotel-internet','hotel-type','hotel-parking'] |
| 'hotel-stars' | ['hotel-book day','hotel-name','hotel-book stay','hotel-pricerange','hotel-area', 'hotel-book people','hotel-internet','hotel-type','hotel-parking'] |
| 'hotel-type' | ['hotel-book day','hotel-book stay','hotel-pricerange','hotel-stars','hotel-area', 'hotel-book people','hotel-internet','hotel-parking'] |
| 'hotel-name' | ['hotel-book day','hotel-book stay','hotel-pricerange','hotel-stars','hotel-area', 'hotel-book people','hotel-internet','hotel-parking'] |
| 'hotel-book day' | ['hotel-name','hotel-book stay','hotel-pricerange','hotel-stars','hotel-area', 'hotel-book people','hotel-internet','hotel-type','hotel-parking'] |
| 'hotel-book people' | ['hotel-book day','hotel-name','hotel-book stay','hotel-pricerange','hotel-stars', 'hotel-area','hotel-internet','hotel-type','hotel-parking'] |
| 'hotel-book stay' | ['hotel-book day','hotel-name','hotel-pricerange','hotel-stars','hotel-area', 'hotel-book people','hotel-internet','hotel-type','hotel-parking'] |
| 'restaurant-area' | ['restaurant-book day','restaurant-name','restaurant-food','restaurant-book people', 'restaurant-book time','restaurant-pricerange'] |
| 'restaurant-food' | ['restaurant-book day','restaurant-book people','restaurant-book time', 'restaurant-area','restaurant-pricerange'] |
| 'restaurant-pricerange' | ['restaurant-book day','restaurant-name','restaurant-food','restaurant-book people', 'restaurant-book time','restaurant-area'] |
| 'restaurant-name' | ['restaurant-book day','restaurant-book people','restaurant-book time', 'restaurant-area','restaurant-pricerange'] |
| 'restaurant-book day' | ['restaurant-name','restaurant-food','restaurant-book people','restaurant-book time', 'restaurant-area','restaurant-pricerange'] |
| 'restaurant-book people' | ['restaurant-book day','restaurant-name','restaurant-food','restaurant-book time', 'restaurant-area','restaurant-pricerange'] |
| 'restaurant-book time' | ['restaurant-book day','restaurant-name','restaurant-food','restaurant-book people', 'restaurant-area','restaurant-pricerange'] |
| 'taxi-departure' | ['taxi-destination', 'taxi-leaveat', 'taxi-arriveby'] |
| 'taxi-destination' | ['taxi-departure', 'taxi-leaveat', 'taxi-arriveby'] |
| 'taxi-leaveat' | ['taxi-departure', 'taxi-destination', 'taxi-arriveby'] |
| 'taxi-arriveby' | ['taxi-departure', 'taxi-destination', 'taxi-leaveat'] |
| 'train-arriveby' | ['train-book people','train-day','train-leaveat','train-departure','train-destination'] |
| 'train-leaveat' | ['train-book people','train-arriveby','train-day','train-departure','train-destination'] |
| 'train-departure' | ['train-book people','train-arriveby','train-day','train-leaveat','train-destination'] |
| 'train-destination' | ['train-book people','train-arriveby','train-day','train-leaveat','train-departure'] |
| 'train-day' | ['train-book people','train-arriveby','train-leaveat','train-departure','train-destination'] |
| 'train-book people' | ['train-arriveby','train-day','train-leaveat','train-departure','train-destination'] |
| 'attraction-name' | ['attraction-area'] |
| 'attraction-area' | ['attraction-type'] |
| 'attraction-type' | ['attraction-area'] |

Table 9: Slot-combination dictionary for *neu* case.

| slot-name | *rare* |
|---|---|
| 'hotel-internet' | ['hotel-book people','hotel-book day','hotel-name','hotel-book stay'] |
| 'hotel-area': | ['hotel-book people','hotel-book day','hotel-name','hotel-book stay'] |
| 'hotel-parking' | ['hotel-book people','hotel-book day','hotel-name','hotel-book stay'] |
| 'hotel-pricerange' | ['hotel-book people','hotel-book day','hotel-name','hotel-book stay'] |
| 'hotel-stars' | ['hotel-book people','hotel-book day','hotel-name','hotel-book stay'] |
| 'hotel-type' | ['hotel-book people','hotel-book day','hotel-book stay'] |
| 'hotel-name' | ['hotel-pricerange','hotel-stars','hotel-area','hotel-internet','hotel-parking'] |
| 'hotel-book day' | ['hotel-name','hotel-pricerange','hotel-stars','hotel-area','hotel-internet', 'hotel-type','hotel-parking'] |
| 'hotel-book people' | ['hotel-name','hotel-pricerange','hotel-stars','hotel-area','hotel-internet', 'hotel-type','hotel-parking'] |
| 'hotel-book stay' | ['hotel-name','hotel-pricerange','hotel-stars','hotel-area','hotel-internet', 'hotel-type','hotel-parking'] |
| 'restaurant-area' | ['restaurant-book day','restaurant-name','restaurant-book time', 'restaurant-book people'] |
| 'restaurant-food' | ['restaurant-book day','restaurant-book time','restaurant-book people'] |
| 'restaurant-pricerange' | ['restaurant-book day','restaurant-name','restaurant-book time', 'restaurant-book people'] |
| 'restaurant-name' | ['restaurant-area','restaurant-pricerange'] |
| 'restaurant-book day' | ['restaurant-name','restaurant-area','restaurant-food','restaurant-pricerange'] |
| 'restaurant-book people' | ['restaurant-name','restaurant-area','restaurant-food','restaurant-pricerange'] |
| 'restaurant-book time' | ['restaurant-name','restaurant-area','restaurant-food','restaurant-pricerange'] |
| 'taxi-departure' | [] |
| 'taxi-destination' | [] |
| 'taxi-leaveat' | ['taxi-departure', 'taxi-destination'] |
| 'taxi-arriveby' | ['taxi-departure', 'taxi-destination'] |
| 'train-arriveby' | ['train-destination', 'train-departure'] |
| 'train-leaveat' | ['train-destination','train-book people','train-arriveby','train-departure'] |
| 'train-departure' | [] |
| 'train-destination' | [] |
| 'train-day' | ['train-destination', 'train-departure'] |
| 'train-book people' | ['train-arriveby','train-departure','train-destination','train-day','train-leaveat'] |
| 'attraction-name' | ['attraction-area'] |
| 'attraction-area' | ['attraction-name'] |
| 'attraction-type' | [] |

Table 10: Slot-combination dictionary for *rare* case.

# I    SLOT-VALUE DICTIONARY

Please find the different slot-value dictionaries introduced in the main paper below.

| slot-name | train-$O$ |
|---|---|
| "hotel-internet" | ['yes'] |
| "hotel-type" | ['hotel', 'guesthouse'] |
| "hotel-parking" | ['yes'] |
| "hotel-pricerange": | ['moderate', 'cheap', 'expensive'] |
| "hotel-book day" | ["march 11th", "march 12th", "march 13th", "march 14th", "march 15th", "march 16th", "march 17th","march 18th", "march 19th", "march 20th"] |
| "hotel-book people" | ["20","21","22","23","24","25","26","27","28","29"] |
| "hotel-book stay" | ["20","21","22","23","24","25","26","27","28","29"] |
| "hotel-area" | ['south', 'north', 'west', 'east', 'centre'] |
| "hotel-stars" | ['0', '1', '2', '3', '4', '5'] |
| "hotel-name" | ["moody moon", "four seasons hotel", "knights inn", "travelodge", "jack summer inn", "paradise point resort"] |
| "restaurant-area" | ['south', 'north', 'west', 'east', 'centre'] |
| "restaurant-food" | ['asian fusion', 'burger', 'pasta', 'ramen', 'taiwanese'] |
| "restaurant-pricerange": | ['moderate', 'cheap', 'expensive'] |
| "restaurant-name" | ["buddha bowls","pizza my heart","pho bistro", "sushiya express","rockfire grill","itsuki restaurant"] |
| "restaurant-book day" | ["monday","tuesday","wednesday","thursday","friday", "saturday","sunday"] |
| "restaurant-book people" | ["20","21","22","23","24","25","26","27","28","29"] |
| "restaurant-book time": | ["19:01","18:06","17:11","19:16","18:21","17:26","19:31", "18:36","17:41","19:46","18:51","17:56", "7:00 pm", "6:07 pm","5:12 pm","7:17 pm","6:17 pm","5:27 pm", "7:32 pm","6:37 pm","5:42 pm","7:47 pm","6:52 pm", "5:57 pm", "11:00 am","11:05 am","11:10 am","11:15 am", "11:20 am","11:25 am","11:30 am","11:35 am","11:40 am", "11:45 am","11:50 am", "11:55 am"] |
| "taxi-arriveby" | [ "17:26","19:31","18:36","17:41","19:46","18:51","17:56", "7:00 pm","6:07 pm","5:12 pm","7:17 pm","6:17 pm", "5:27 pm","11:30 am","11:35 am","11:40 am","11:45 am", "11:50 am","11:55 am"] |
| "taxi-leaveat": | [ "19:01","18:06","17:11","19:16","18:21","7:32 pm", "6:37 pm","5:42 pm","7:47 pm","6:52 pm", "5:57 pm","11:00 am","11:05 am","11:10 am", "11:15 am","11:20 am","11:25 am"] |
| "taxi-departure": | ["moody moon", "four seasons hotel", "knights inn", "travelodge", "jack summer inn", "paradise point resort"], |
| "taxi-destination": | ["buddha bowls","pizza my heart","pho bistro", "sushiya express","rockfire grill","itsuki restaurant"] |
| "train-arriveby": | [ "17:26","19:31","18:36","17:41","19:46","18:51", "17:56","7:00 pm","6:07 pm","5:12 pm","7:17 pm", "6:17 pm","5:27 pm","11:30 am","11:35 am","11:40 am", "11:45 am","11:50 am","11:55 am"] |
| "train-leaveat": | ["19:01","18:06","17:11","19:16","18:21", "7:32 pm", "6:37 pm","5:42 pm","7:47 pm","6:52 pm","5:57 pm", "11:00 am","11:05 am","11:10 am","11:15 am","11:20 am","11:25 am"] |
| "train-departure" | ["gilroy","san martin","morgan hill","blossom hill", "college park","santa clara","lawrence","sunnyvale"] |
| "train-destination" | ["mountain view","san antonio","palo alto","menlo park", "hayward park","san mateo","broadway","san bruno"] |
| "train-day": | ["march 11th", "march 12th", "march 13th", "march 14th", "march 15th", "march 16th", "march 17th","march 18th", "march 19th", "march 20th"] |
| "train-book people" | ["20","21","22","23","24","25","26","27","28","29"] |
| "attraction-area" | ['south', 'north', 'west', 'east', 'centre'] |
| "attraction-name" | ["grand canyon","golden gate bridge","niagara falls", "kennedy space center","pike place market","las vegas strip"] |
| "attraction-type" | ['historical landmark', 'aquaria', 'beach', 'castle','art gallery'] |

Table 11: Slot value dictionary of train-$O$.

| slot-name | $I$ |
|---|---|
| "hotel-internet" | ['yes'] |
| "hotel-type" | ['hotel', 'guesthouse'] |
| "hotel-parking" | ['yes'] |
| "hotel-pricerange" | ['moderate', 'cheap', 'expensive'] |
| "hotel-book day" | ['friday', 'tuesday', 'thursday', 'saturday', 'monday', 'sunday', 'wednesday'] |
| "hotel-book people" | ['1', '2', '3', '4','5', '6', '7','8'] |
| "hotel-book stay" | ['1', '2', '3', '4','5', '6', '7','8'] |
| "hotel-name" | ['alpha milton', 'flinches bed and breakfast', 'express holiday inn by cambridge', 'wankworth house', 'alexander b and b', 'the gonville hotel'] |
| "hotel-stars" | ['0', '1', '3', '2', '4', '5'] |
| "hotel-area" | ['south', 'east', 'west', 'north', 'centre'] |
| "restaurant-area" | ['south', 'east', 'west', 'north', 'centre'] |
| "restaurant-food" | ['europeon', 'brazliian', 'weish'] |
| "restaurant-pricerange" | ['moderate', 'cheap', 'expensive'] |
| "restaurant-name": | ['pizza hut in cherry', 'the nirala', 'barbakan', 'the golden house', 'michaelhouse', 'bridge', 'varsity restaurant','loch', 'the peking', 'charlie', 'cambridge lodge', 'maharajah tandoori'] |
| "restaurant-book day" | ['friday', 'tuesday', 'thursday', 'saturday', 'monday', 'sunday', 'wednesday'] |
| "restaurant-book people" | ['8', '6', '7', '1', '3', '2', '4', '5'] |
| "restaurant-book time" | ['14:40', '19:00', '15:15', '9:30', '7 pm', '11 am', '8:45'] |
| "taxi-arriveby" | ['08:30', '9:45'] |
| "taxi-leaveat" | ['7 pm', '3:00'] |
| "taxi-departure" | ['aylesbray lodge', 'fitzbillies', 'uno', 'zizzi cambridge', 'express by holiday inn', 'great saint marys church', 'county folk museum','riverboat', 'bishops stortford', 'caffee uno', 'hong house', 'gandhi', 'cambridge arts', 'the hotpot', 'regency gallery', 'saint johns chop shop house'] , |
| "taxi-destination" | ['ashley', 'all saints', "de luca cucina and bar's", 'the lensfield hotel', 'oak bistro', 'broxbourne', 'sleeperz hotel', "saint catherine's college"] |
| "train-arriveby" | ['4:45 pm', '18:35', '21:08', '19:54', '10:08', '13:06', '15:24', '07:08', '16:23', '8:56', '09:01', '10:23', '10:00 am', '16:44', '6:15', '06:01', '8:54','21:51', '16:07', '12:43', '20:08', '08:23', '12:56', '17:23', '11:32', '20:54', '20:06', '14:24', '18:10', '20:38', '16:06', '3:00', '22:06', '20:20', '17:51','19:52', '7:52', '07:44', '16:08'], |
| "train-leaveat" | ['13:36', '15:17', '14:21', '3:15 pm', '6:10 am', '14:40', '5:40', '13:40', '17:11', '13:50', '5:11', '11:17', '5:01', '13:24', '5:35', '07:00', '8:08', '7:40', '11:54', '12:06', '07:01', '18:09', '13:17', '21:45', '06:40', '01:44', '9:17', '20:21', '20:40', '08:11', '07:35', '14:19', '1 pm', '19:17', '19:48', '19:50', '10:36', '09:19', '19:35', '8:06', '05:29', '17:50', '15:16', '09:17', '7:35', '5:29', '17:16', '14:01', '10:21', '05:01', '15:39', '15:01', '10:11', '08:01'], |
| "train-departure": | ['london liverpool street', 'kings lynn', 'norwich', 'birmingham new street', 'london kings cross','broxbourne'] |
| "train-destination" | ['bishops stortford', 'cambridge', 'ely', 'stansted airport', 'peterborough', 'leicester', 'stevenage'] |
| "train-day" | ['friday', 'tuesday', 'thursday', 'monday', 'saturday', 'sunday', 'wednesday'] |
| "train-book people" | ['9'] |
| "attraction-name": | ['the cambridge arts theatre', 'the churchill college', 'the castle galleries', 'cambridge', "saint catherine's college", 'street', 'corn cambridge exchange', 'fitzwilliam', 'cafe jello museum'], |
| "attraction-area": | ['south', 'east', 'west', 'north', 'centre'], |
| "attraction-type" | ['concerthall', 'museum', 'entertainment', 'college', 'multiple sports', 'hiking', 'architecture', 'theatre', 'cinema', 'swimmingpool', 'boat', 'nightclub', 'park'] |

Table 12: Slot-value dictionary for $I$ case.

| slot-name | $O$ |
|---|---|
| "hotel-internet" | ['yes'] |
| "hotel-type" | ['hotel', 'guesthouse'] |
| "hotel-parking" | ['yes'] |
| "hotel-pricerange" | ['moderate', 'cheap', 'expensive'] |
| "hotel-book day" | ["april 11th", "april 12th", "april 13th", "april 14th", "april 15th", "april 16th", "april 17th","april 18th", "april 19th", "april 20th"] |
| "hotel-book people" | ["30","31","32","33","34","35","36","37","38","39"] |
| "hotel-book stay" | ["30","31","32","33","34","35","36","37","38","39"] |
| "hotel-area" | ['south', 'east', 'west', 'north', 'centre'] |
| "hotel-stars" | ['0', '1', '2', '3', '4', '5'] |
| "hotel-name" | ["white rock hotel", "jade bay resort", "grand hyatt", "hilton garden inn" ,"cottage motel","mandarin oriental"], |
| "restaurant-area" | ['south', 'east', 'west', 'north', 'centre'] |
| "restaurant-food" | ["sichuan", "fish", "noodle", "lobster", "burrito", "dumpling", "curry","taco"] |
| "restaurant-pricerange" | ['moderate', 'cheap', 'expensive'] |
| "restaurant-name": | ["lure fish house","black sheep restaurant","palapa restaurant", "nikka ramen", "sun sushi","super cucas"] |
| "restaurant-book day": | ["monday","tuesday","wednesday","thursday","friday","saturday","sunday"] |
| "restaurant-book people" | ["30","31","32","33","34","35","36","37","38","39"] |
| "restaurant-book time" | ["20:02","21:07","22:12","20:17","21:22","22:27","20:32","21:37","22:42", "20:47","21:52","22:57","8:00 pm","9:04 pm","10:09 pm","8:14 pm", "9:19 pm","10:24 pm","8:29 pm","9:34 pm","10:39 pm","8:44 pm","9:49 pm", "10:54 pm","10:00 am","10:06 am","10:11 am","10:16 am","10:21 am","10:26 am", "10:31 am","10:36 am","10:41 am","10:46 am","10:51 am","10:56 am"], |
| "taxi-arriveby": | ["20:02","21:07","22:12","20:17","21:22","22:27","9:34 pm","10:39 pm", "8:44 pm","9:49 pm","10:54 pm", "10:00 am","10:06 am","10:11 am", "10:16 am","10:21 am","10:26 am"], |
| "taxi-leaveat": | ["21:37","22:42","20:47","21:52","22:57","8:00 pm","9:04 pm","10:09 pm", "8:14 pm","9:19 pm","10:24 pm","8:29 pm","10:31 am","10:36 am", "10:41 am", "10:46 am","10:51 am","10:56 am"], |
| "taxi-departure": | ["lure fish house","black sheep restaurant","palapa restaurant", "nikka ramen", "sun sushi","super cucas"], |
| "taxi-destination": | ["white rock hotel", "jade bay resort", "grand hyatt", "hilton garden inn", "cottage motel","mandarin oriental"] |
| "train-departure" | ["northridge","camarillo","oxnard","morepark","simi valley","chatsworth", "van nuys","glendale"] |
| "train-destination" | ["norwalk","buena park","fullerton","santa ana","tustin","irvine", "san clemente","oceanside"], |
| "train-arriveby": | ["20:02","21:07","22:12","20:17","21:22","22:27", "9:34 pm","10:39 pm", "8:44 pm","9:49 pm","10:54 pm","10:00 am","10:06 am","10:11 am", "10:16 am","10:21 am","10:26 am"], |
| "train-day": | ["april 11th", "april 12th", "april 13th", "april 14th", "april 15th", "april 16th", "april 17th","april 18th", "april 19th", "april 20th"], |
| "train-leaveat": | ["21:37","22:42","20:47","21:52","22:57","8:00 pm","9:04 pm","10:09 pm", "8:14 pm","9:19 pm","10:24 pm","8:29 pm","10:31 am","10:36 am", "10:41 am","10:46 am","10:51 am","10:56 am"], |
| "train-book people": | ["30","31","32","33","34","35","36","37","38","39"] |
| "attraction-area": | ['south', 'east', 'west', 'north', 'centre'] |
| "attraction-name": | ["statue of liberty","empire state building","mount rushmore", "brooklyn bridge","lincoln memorial","times square"], |
| "attraction-type": | ["temple", "zoo", "library", "skyscraper","monument"] |

Table 13: Slot-value dictionary for $O$ case.

