# OpenReview forum: "CoCo: Controllable Counterfactuals for Evaluating Dialogue State Trackers"
_ICLR.cc/2021/Conference — ICLR 2021 Poster_

### Official Review · AnonReviewer2 · 2020-10-28
**Creating challenging examples for DST systems: interesting results but may need more clarification in the methods section**

**Rating:** 6
**Confidence:** 3

**Review:**

Update after reading author response:

Thank you for the thoughtful response.  I appreciate that you have added extra implementation details and am changing my score to a 6.

Regarding the limitations in scope:  My apologies if my review was confusingly worded.  I just wanted to clarify that I had meant that the counterfactual generation method itself may have limitations (not the high-level DST task, which I agree has broad uses).  The concern is that the adversarial example generation strategy might be too domain-specific to transfer easily to other tasks, and that this might limit the impact of the proposed counterfactual generation method.  I think this is somewhat related to the parts of the methodology that R1 described as "ad-hoc", "engineering intensive", or reliant on heuristics.  I still have some reservations about the transferability of the proposed methods, though the authors' response to R1 did clarify a bit on this point.

-------------------------------------------------------------------------------------------------------------------------------------------

Original Review:

This paper presents CoCo, a new method for generating adversarial examples for DST tasks using “controllable counterfactual” generation.  Unlike other approaches for adversarial example generation, this approach is model agnostic.  They also demonstrate the effectiveness of the method by applying the Multi-woz dataset where s.o.t.a. Model performance drops by ~30 points.

I might lean towards rejecting this paper because there are several points in the methods that are still unclear to me.  However, I hope the authors may be able to clarify some of my questions (I’ve listed a few in the limitations section of my review).  Another potential drawback is that the methods described here are limited in scope to the dst domain, so it may not be as useful to other ML researchers more broadly.

Strengths:
- CoCo is a very effective method.  Model performance drops by about 30 points, demonstrating that the adversarial examples generated are extremely challenging.
- The human study demonstrates that the created examples are still human-like and are actually often rated “more correct” than the original Multi-Woz responses.

Limitations:
- The proposed methods are using a lot of pre-existing components and limited in scope to this one domain.  While the created counterfactuals are useful as a challenge dataset for DST systems, the overall approach in this paper may not be more broadly impactful outside of this task.
- There are a few points in the methodology that seem a bit unclear or should be described more fully:
  - Can you provide more details about the conditional generation model (p):  How is it being trained?  What architecture is being used?
  - The Slot-Value match filter: How are you judging whether the candidate “contains” the value? (Is this exact match?)
  - Classifier filter:  Can you provide more details on the classifier architecture and how it is being trained?  What is the precision and recall of its predictions?

Questions:
- In section 5.4: Any thoughts on why data augmentation is more effective for the Trippy model than the other two models?

Minor Feedback:
- The CoCo name could be easily confused with Microsoft Coco, a commonly used computer vision dataset.  Authors might consider changing the name to avoid confusion.
- The font in Figures 3 and 4 is tiny and hard to read.  I recommend making it bigger.

---

> ### Author Response · Authors · 2020-11-21
> **Thank you for your review**
>
> Thank you for your comments! We revised the paper to include more details based on your feedback. Please find our responses to your comments below:
>
> **Limitation 1- The overall approach in this paper may not be more broadly impactful outside of DST task.**
>
> A significant portion of dialogue research focuses on DST task, which also attracts much attention from the industry in recent years. We believe DST task itself has been a broad enough research area, being the most critical module of task-oriented dialogue systems.  Hence, our work has the potential to inspire many future works in this domain. In addition, our proposed approach can improve SOTA DST performance by 1.3% as a side product although it is not the main focus of this work,. Our experiments for a follow-up work show that augmenting training data with 8x CoCo-generated conversations can improve SOTA DST models by absolute 5% compared to 1.3% improvement with 1x augmentation as reported in this paper.
>
> Besides, our  approach can generate very high-quality conversation turns verified by human evaluations, and we think that it may also be applicable to the following structured controllable text generation problems beyond the DST domain:
>
> (1) Controllable sentence-level text generation. Our approach might be useful for synthesizing novel and high-quality sentences by modifying and conditioning on other structured representations (e.g., syntactic parse tree, AMR).
>
> (2) Controllable table-to-text generation. One can train an encoder-decoder model from table entries into natural language, which then can be fed with modified entries or even a new table during inference to generate the corresponding text summarizing the table.
>
> (3) Controllable Text Summarization. [ICLR 2021, Under Review, https://openreview.net/forum?id=ohdw3t-8VCY]. This paper proposes to train an encoder-decoder model for document summarization tasks with key phrases as an additional input of the encoder.  During inference, users can change key phrases and the model can generate summaries only including these key phrases.
>
> **Limitation 2-(1) Details about the conditional generation model (p): How is it being trained? What architecture is being used?**
>
> Our conditional generation model is an encoder-decoder architecture. We initialize p with a pre-trained T5-small model [1] and use the cross-entropy loss to maximize the likelihood of the target sequence conditioned on the concatenation of conversation history and turn-level belief state, which is flattened into a single sequence using appropriate special separator tokens as illustrated in Fig.2. We also reflected these by revising Fig.2 and Section 4.2, now including more details about the model and how it is trained. Please also see Appendix D.1 for further details on our hyperparameter selection for the conditional generation model.
>
> **Limitation 2-(2) The Slot-Value match filter: How are you judging whether the candidate “contains” the value?**
>
> Yes, we use exact string matches to eliminate utterance candidates. Please see the “Slot-Value Match Filter” part in Section 4.3 for more details.
>
> **Limitation 2-(3) Classifier filter: Can you provide more details on the classifier architecture and how it is being trained? What is the precision and recall of its predictions?**
>
> We instantiate our classifier filter with a BERT-base uncased model. We feed the hidden representation of  [CLS] from BERT into a linear layer with an output size of N (number of slots) followed by a sigmoid function, whose output has N dimensions and each of them represents the probability that the corresponding slot appears in the last user turn given the conversation history. We use binary cross-entropy loss to train our classifier filter. The classifier achieves 92.3% of precision and 93.5% of recall on the development set. We revised the “Classifier Filter” part in Section 4.3 to include more details on the classifier architecture along with its training objective and the prediction performance. Please also see Appendix D.2 for further details on our hyperparameter selection for the classifier.
>
> **Question 1: why data augmentation is more effective for the Trippy model than the other two models?**
>
> Our hypothesis is that the Triple-copy mechanism in TripPy has a much smaller search space than other two generative models. Even without data augmentation, TripPy is also the most robust model compared to other two models as shown in  Fig. 3.
>
> **Minor Feedback 1-The CoCo name could be confused with Microsoft Coco and can be changed into another name.**
>
> We agree with the reviewer on the potential confusion that might arise out of the naming. We plan to change its name to “CoCo-DST” for the camera-ready version.
>
> **Minor Feedback 2-The font in Figures 3 and 4 is tiny and hard to read.**
>
> We have enlarged the fonts in  Fig. 3 and 4 in the new version.
>
> REFERENCES
>
> [1] Exploring the Limits of Transfer Learning with Unified Text-to-Text Transformer.

---

### Official Review · AnonReviewer1 · 2020-10-28
**Review #1**

**Rating:** 4
**Confidence:** 5

**Review:**

<Summary>

This paper addresses the problem of evaluating dialogue state trackers (DST)’s generalization ability to novel and realistic dialogue scenarios that do not exist in the dataset. It proposes a model-independent approach to evaluate DST systems with the idea of counterfactual conversation generation. The proposed approach is integrated with three recent DST models and evaluated on MultiWoZ dataset.

<Strengths>

1. The idea of counterfactual goal/conversation generation can be useful for the evaluation of DST systems, which is often quite challenging in practice.

2. The proposed CoCo idea is tested with three recent DST systems including Trad, TripPy and SimpleTod and show its effectiveness on MultiWoZ dataset.

<Weakness>

1. This paper proposes an interesting idea for DST evaluation but its implementation is largely ad-hoc and engineering intensive and thus bears little technical novelty.

(1) As described in section 4 and Fig.2, the proposed approach is simply a combination of multiple components with heuristic rules (e.g. the way of value substitution, and the use of some predefined operations).

(2) The two filtering schemes are also ad-hoc, consisting of two slot-value match filter and classifier fitter.

(3) It is hard to find methodological novelty in the proposed method. Given that ICLR is a top premier ML venue, it could be a significant weakness to be a publishable work. Thus, this work may be better fit to a venue of NLP.

2. Experimental evaluation can be improved.

(1) This work evaluates only on a single dataset – MultiWoZ 2.1. Admittedly, it is one of the most important benchmarks for DST tests, but one or more datasets are encouraged to use for better justification of generality, for example, MultiWoZ 2.0 or Taskmaster-1.

(2) The experiments mainly focus on how much the proposed CoCo improves the three DST systems. However, comparative study is required such as how much CoCo is better than other alternatives or baselines.

<Conclusion>

My initial decision is ‘reject’ mainly due to lack of technical novelty. Experiments could be improved for better evaluation.

---

> ### Author Response · Authors · 2020-11-21
> **Thank you for your review**
>
> Thank you for your comments! Please find our responses below:
>
> **W1-(1) the proposed approach is simply a combination of multiple components with heuristic rules.**
>
> We agree that some designs on the counterfactual goal generation are based on heuristic rules. However, it is not clear why employing certain heuristics should necessarily be a weakness, especially when they are under reasonable assumptions and serve as part of a greater solution effectively addressing an important problem and making useful conclusions on the potential issues with evaluation of SOTA DST models.
>
> *Revisiting our approach to clarify and share exactly what part of it involves heuristic decisions:*
>
> Our proposed approach consists of two main components: counterfactual goal generation and counterfactual conversation generation. Counterfactual-goal generation, as illustrated in Fig. 2, has three operations (e.g., drop, change, and add) that are applied on the original turn-level belief state to derive a counterfactual goal, which stands as the control engine behind the user utterance generation. However, simply sampling a turn-level belief state from the entire space often breaks the dialog-flow due to its potential conflict with the conversation history. This is the only part in the whole pipeline where some heuristic rules are designed to constrain the sampling space so that new goals are consistent with the dialog flow. Counterfactual conversation generation, on the other hand, is a completely novel way of generating user utterances using turn-level belief states to control the user intent. This structured representation is encoded together with the conversation history, which is then fed to decoder to generate a user utterance reflecting their goal. Finally, we employ a novel filtering mechanism to eliminate the undesirable utterances that fail to reflect the user goal.
>
> **W1-(2) The two filtering schemes are also ad-hoc.**
>
> We decompose the source of undesirable utterances into two categories: degeneration and over-generation. To tackle the first, we use the most natural strategy to filter the utterances that miss a slot value from the counterfactual goal. However, it cannot eliminate the utterances that hallucinate a slot that does not exist in the counterfactual goal. To tackle it, we propose a classifier-based filtering that predicts what slots appear in the dialogue turn, and eliminates the utterances with additional slots outside of the goal. We argue that the combined filtering mechanism is simple, but very effective as shown by the correctness metric in human evaluation.
>
> **W1-(3) Hard to find methodological novelty in the proposed method.**
>
> In this work, we identify an important problem, which is also acknowledged by the reviewer, for which we propose a simple yet effective solution with strong results not only clearly showing the need for more comprehensive evaluation but also improving the current SOTA by 1.3% as a side product. Although heuristic decisions are within our approach to preserve the dialogue consistency, we respectfully disagree with the reviewer on his comment regarding the lack of novelty.
>
> **Novelty of the paper**
>
> Collecting a large-scale task-oriented dataset is expensive and time-consuming [1]. It is not feasible to cover the possible conversations in real-world via human collected dialogues. To this end, we propose a principled, and model-agnostic method to evaluate these models beyond human annotation. Our approach is the first step to bridge this gap by generating human-like conversations controlled by their underlying goals. We also clearly contrast this paper with prior work and state its differences in the 3rd paragraph of the Introduction section to better isolate its novelty.
>
> **W2-(1) This work evaluates only on a single dataset – MultiWoZ 2.1.**
>
> We report our results on MultiWoZ 2.1 because (i) It is more widely used than MultiWoZ 2.0 by the recent DST works (see https://github.com/budzianowski/multiwoz), and (ii) We observe similar results on MultiWoZ 2.0 and didn’t report them to avoid redundancy, and finally (iii) We do not conduct experiments on TaskMaster dataset as none of the SOTA DST models report results on it.
>
> **W2-(2) The experiments mainly focus on how much the proposed CoCo improves the three DST systems...**
>
> Our main focus is to propose a principled and model-agnostic method to evaluate DST models beyond held-out set accuracy. We explored the effectiveness of CoCo as a data augmentation method in improving DST models only in subsection (5.4), at the end of which we hint that it is worth exploring further and leave it as future work. Our experiments in a follow-up work show that augmenting training data with 8x CoCo-generated conversations can improve SOTA DST models by absolute 5% compared to 1.3% with 1x augmentation reported in this paper.
>
> REFERENCES:
>
> [1] MultiWOZ - A Large-Scale Multi-Domain Wizard-of-Oz Dataset for Task-Oriented Dialogue Modelling. EMNLP’18.

---

### Official Review · AnonReviewer4 · 2020-10-28
**Nice work on generating conversations to evaluate models for Dialog State Tracking**

**Rating:** 7
**Confidence:** 4

**Review:**

This paper presents an interesting approach to generate dialogs in a controllable fashion to evaluate a Dialog State Tracking system on a data distribution which is different from the training/test data. The proposed approach first generates a turn-level goal by adding or dropping a slot and then replacing slot values. In the second step, the proposed method generates counterfactual conversation conditioned on the dialog history and goal generated in the previous step. The authors show that evaluating current state-of-the-art DST model on MultiWOZ datasets with the generated counterfactuals results in significant performance drop. Additionally, human evaluation shows that the generated conversations perfectly reflect the underlying user goal.

The paper is trying to tackle an important problem of evaluating robustness of a DST model when most of the available datasets has similar distribution in train and test splits. The proposed method is well explained and the effectiveness of the approach is substantiated by extensive results.

It is not clear what is the performance of utterance generation model. This model is somewhat different from other language model since it is conditioned on belief as well. Also, seems like the proposed approach can generate counterfactual conversation only for one turn which seems limited for the evaluation.

I would like to hear from authors on:
- In section 5.4, it is not clear if slot combination dictionary was also split such that slot combination in train and test data doesn’t overlap? If not, model can learn just the generated utterance pattern and perform better on the generated test set.
- Performance of the utterance generation model in terms of diversity of the generated utterances since it will affect the evaluation’s robustness.

---

> ### Author Response · Authors · 2020-11-21
> **Thank you for your review**
>
> Thank you for your feedback! Please find our responses to your questions below:
>
> **Q1: In section 5.4, it is not clear if slot combination dictionary was also split such that slot combination in train and test data doesn’t overlap?**
>
> Fig. 3 shows that all three DST models are consistently most susceptible to conversations generated by COCO+(rare) strategy. That is why we use the “rare” slot combination dictionary and the “train-O” slot value dictionary to augment the training data to retrain models.
> Since the ontology is the same across the train, dev, and test splits of MultiWOZ dataset, any non-empty slot combination dictionary would have to overlap with the test set (please see Table-2 of the revised manuscript).
> In Fig. 4, in comparison with the original models, we present the performance of retrained models on the augmented data (COCO+(rare)) against each generated test set (“freq”, “neu”, “rare”), across which we see similar improvements. This indicates that the improvement provided by the data augmentation is not due to overlapping slot combinations. For example, the test set generated with the “freq” slot-combination dictionary has minimal overlap with “rare” slot-combinations, but they still benefit a lot from the data augmentation by the “rare” slot dictionary.
> Besides, data augmentation with the rare slot combination dictionary is also very effective on the original test (1.3% absolute improvement on TripPy with 1x augmentation as reported in this paper). Our experiments for a follow-up work show that augmenting training data with 8x CoCo-generated conversations can improve the joint goal accuracy of SOTA DST models by 5%. These results indicate that retrained DST models not only improve on data with the same slot combination dictionary in training but also improve on the original test set which in principle has unknown slot combination statistics, though using the same ontology.
>
>
> **Q2: Performance of the utterance generation model in terms of diversity of the generated utterances since it will affect the evaluation’s robustness.**
>
> Based on the reviewer’s question, we add the diversity evaluation of generated user utterances in the Appendix E of the revised manuscript. We analyze the diversity from two perspectives (1) slot-combination diversity and (2) utterance lexical diversity.
> Slot-combination diversity: In Table-5 under Appendix E, we compare the diversity of generated utterances between the original test set (Ori-test) and CoCo generated test set (CoCo-test) in terms of their slot co-occurrence distributions. The distribution entropy of CoCo-test (0.65) is higher than its counterpart of Ori-test (0.57) with the upper bound of 0.78 corresponding to the uniform distribution. It indicates that CoCo-test is even more diverse compared to Ori-test in terms of slot combinations, which in return leads to the generation of more diverse utterances.
> Utterance lexical diversity: We report lexical diversity results in Table-6 of section E in the Appendix. Overall, it shows that CoCo-test has a similar lexical diversity score (Distinct-k [2]) with the original test set of MultiWOZ.
>
>
> Based on these two diversity measures, we conclude that the CoCo-generated utterances are at least as diverse as the original utterances in MultiWOZ, hence it should not negatively affect the evaluation’s robustness.
>
> REFERENCES
>
> [1] Vijayakumar, A.K., Cogswell, M., Selvaraju, R.R., Sun, Q., Lee, S., Crandall, D.J., & Batra, D. (2016). Diverse Beam Search: Decoding Diverse Solutions from Neural Sequence Models. ArXiv, abs/1610.02424.
>
> [2] Diversity Promoting Objective Function for Neural Conversation Models. Li et. al. NAACL’16 (https://www.aclweb.org/anthology/N16-1014.pdf).

---

### Decision · Program_Chairs · 2021-01-07
**Final Decision**

**Decision:**

Accept (Poster)

**Comment:**

The paper proposes a method  to generate conversations for evaluate dialog systems using counterfactual generation.

Pros:
- The reviewers agree that the paper makes a good contribution towards evaluation of DST models.
- The paper adds to a growing body of work on robust evaluation of NLP models

Cons:
- One reviewer has commented on the lack of novelty. However, I believe that the authors have adequately addressed it. In particular, I do not see any harm in using heuristics/templates to generate counterfactuals as long as the final goal of robust evaluation is achieved.
- It would have been good to evaluate the method on other datasets. However, I agree with the authors' rebuttal that this is indeed the most popular dataset for the task and most SOTA methods evaluate on this dataset.

The authors have adequately addressed all reviewer concerns and have clearly highlighted their contributions and novelty.

I think of this as a valuable contribution and would like to see the paper accepted.